# Particulate Matter and Premature Mortality: A Bayesian Meta-Analysis

**DOI:** 10.3390/ijerph18147655

**Published:** 2021-07-19

**Authors:** Nilakshi T. Waidyatillake, Patricia T. Campbell, Don Vicendese, Shyamali C. Dharmage, Ariadna Curto, Mark Stevenson

**Affiliations:** 1Allergy and Lung Health Unit, Melbourne School of Population and Global Health, The University of Melbourne, Melbourne, VIC 3010, Australia; don.vicendese@unimelb.edu.au (D.V.); s.dharmage@unimelb.edu.au (S.C.D.); 2Department of Medical Education, Melbourne Medical School, The University of Melbourne, Melbourne, VIC 3010, Australia; 3Department of Infectious Diseases, Melbourne Medical School, University of Melbourne at the Peter Doherty Institute for Infection and Immunity, Melbourne, VIC 3000, Australia; patricia.campbell@unimelb.edu.au; 4Centre for Epidemiology and Biostatistics, Melbourne School of Population and Global Health, The University of Melbourne, Melbourne, VIC 3010, Australia; 5Department of Mathematics and Statistics, La Trobe University, Bundoora, VIC 3086, Australia; 6Mary MacKillop Institute for Health Research, Australian Catholic University, Melbourne, VIC 3065, Australia; ariadna.curto@isglobal.org; 7Transport Health and Urban Design Research Lab, Melbourne School of Design, The University of Melbourne, Melbourne, VIC 3010, Australia

**Keywords:** Bayesian hierarchical meta-analysis, particulate matter, PM_2.5_, PM_10_, premature mortality

## Abstract

Background: We present a systematic review of studies assessing the association between ambient particulate matter (PM) and premature mortality and the results of a Bayesian hierarchical meta-analysis while accounting for population differences of the included studies. Methods: The review protocol was registered in the PROSPERO systematic review registry. Medline, CINAHL and Global Health databases were systematically searched. Bayesian hierarchical meta-analysis was conducted using a non-informative prior to assess whether the regression coefficients differed across observations due to the heterogeneity among studies. Results: We identified 3248 records for title and abstract review, of which 309 underwent full text screening. Thirty-six studies were included, based on the inclusion criteria. Most of the studies were from China (*n* = 14), India (*n* = 6) and the USA (*n* = 3). PM_2.5_ was the most frequently reported pollutant. PM was estimated using modelling techniques (22 studies), satellite-based measures (four studies) and direct measurements (ten studies). Mortality data were sourced from country-specific mortality statistics for 17 studies, Global Burden of Disease data for 16 studies, WHO data for two studies and life tables for one study. Sixteen studies were included in the Bayesian hierarchical meta-analysis. The meta-analysis revealed that the annual estimate of premature mortality attributed to PM_2.5_ was 253 per 1,000,000 population (95% CI: 90, 643) and 587 per 1,000,000 population (95% CI: 1, 39,746) for PM_10_. Conclusion: 253 premature deaths per million population are associated with exposure to ambient PM_2.5_. We observed an unstable estimate for PM_10_, most likely due to heterogeneity among the studies. Future research efforts should focus on the effects of ambient PM_10_ and premature mortality, as well as include populations outside Asia. Key messages: Ambient PM_2.5_ is associated with premature mortality. Given that rapid urbanization may increase this burden in the coming decades, our study highlights the urgency of implementing air pollution mitigation strategies to reduce the risk to population and planetary health.

## 1. Introduction

Environmental pollution is a global public health problem [1,2]. Despite various preventive strategies, air pollution continues to be a significant contributor to adverse health outcomes, particularly premature mortality [2,3]. Particulate matter (PM) is an important contributor to all air pollutants, with PM_2.5_ and PM_10_ identified as two of the key components. Of the two, PM_2.5_ has been reported to reach into deep tissues, such as lungs, thereby leading to the majority of health-related impacts [4,5]. In the lungs, PM_2.5_ corrodes the alveoli, which may lead to chronic obstructive pulmonary disease (COPD) [5,6]. PM_2.5_ can also lead to peripheral vascular system damage and can directly damage the myocardium leading to arrhythmias, atherosclerosis and stroke [7,8]. The effects of PM_10_ lead to more acute responses, such as wheeze or hyperreactive airways and bronchitis [9]. However, there is evidence that PM_10_ increases cardiovascular mortality [10]. Taken together, evidence indicates that PM may increase the risk of cardio-respiratory morbidity and mortality [11].

There is a growing body of literature on the role of PM in premature mortality [3,12]. Controversy surrounds this area, in part because no synthesis of the evidence has been undertaken that specifically accounts for inconsistencies among studies, especially study population differences. Hence, to date, no research has assessed the association between PM and premature mortality adjusting for the potential influence of the heterogeneity of the findings across multiple studies. This may have led to a biased estimation of health impacts due to PM exposure [13].

In the hierarchy of evidence, randomized controlled trials are the preferred research design upon which to generate evidence. Given it is difficult to apply such methods to the study of air pollution, almost all studies apply observational approaches; such studies have limitations making it difficult to draw precise inferences. Synthesizing evidence from multiple observational studies, however, can strengthen the conclusions that can be drawn. In this study we aim to systematically review the available evidence on PM and its impact on the years of life lost, measured as premature mortality. We also conduct a Bayesian hierarchical meta-analysis to account for the likely heterogeneity between the studies selected for review.

## 2. Methods

Medline, CINAHL and Global Health electronic databases were systematically searched (last accessed January 2020) using keywords and Boolean/phrase terms based on particulate matter and premature mortality (Appendix A: Search strategy). The search was augmented from the reference lists of the included articles. The review was registered in the International Prospective Register of Systematic Reviews (PROSPERO), systematic review registry (CRD42019134760). The inclusion criteria of our systematic review were:Studies that measured PM_2.5_ or PM_10_;Outcome measured as premature mortality;Studies based on any study design;From any population group (no ethnic groups were excluded);Published in English in a peer reviewed journal;Available in Medline, CINAHL and Global Health electronic databases from inception to January 2020.

The exclusion criteria were:Studies which assessed pollutants other than PM_2.5_ and PM_10,_ or measured these pollutants in combination with other pollutants;Literature reviews;Conference papers, abstracts and editorials.

For the purpose of the Bayesian hierarchical meta-analysis, we also included three further selection criteria, namely, (i) Studies that showed log normality of data; (ii) Studies that did not derive PM based on satellite observations; and (iii) Studies providing point estimates with 95% confidence intervals. We excluded studies using solely satellite observations due to the uncertainties linked to satellite-based PM, namely, poor satellite coverage in specific regions, cloud contamination and year-to-year variability as such observations can substantially impact the estimates of premature mortality when compared to global models [14].

Two authors independently reviewed study titles and abstracts for detailed review of the full text (NTW and AC). All duplicates were removed after the initial search. Any disagreements were resolved by consulting with a third, senior author (MS). Studies were excluded after full-text review if they did not meet the inclusion criteria. Data extracted for analysis included the first author’s name, publication year, country, exposure estimates and the method of exposure ascertainment, outcome definitions, the method of outcome ascertainment and key results.

The working definitions for exposures was PM. PM is the particle pollutant component in the atmosphere and is a mixture of solid particles and liquid droplets that can only be seen microscopically. Based on the size of the particles, PM is categorized as PM_10_ and PM_2.5_, defined as follows:PM_10_: inhalable particles, with diameters that are 10 micrometers and smaller; andPM_2.5_: fine inhalable particles, with diameters that are 2.5 micrometers and smaller.

Our outcome, premature mortality, was defined as death that occurs before the average age of death in the specific population group. It is defined as potential years of life lost.

Quality of the included studies: We assessed study quality by using the Newcastle–Ottawa scale (NOS) for observational studies [15]. This scale is comprised of three elements:(i)Four stars are allocated to study group selection (the first element);(ii)Two stars are allocated to comparability of the groups (the second element); and(iii)Three stars are allocated to ascertainment of the exposure and outcome (the final element).

The NOS score ranges from 0–9 and a methodologically robust paper can achieve a total of nine stars; a perfect score. Based on the total number of stars achieved, a study was categorized as good (a total of seven or more stars), fair (five or six stars) or poor (four stars or less) quality.

Statistical analysis: We conducted a Bayesian hierarchical meta-analysis [16,17]. We conducted two analyses; a meta-analysis for PM_2.5_ and a meta-analysis for PM_10_, in which we used as a non-informative prior an improper uniform distribution over the positive real number line, followed by a heterogeneity analysis. The basic steps followed were (i) Checking for log normality of the data; (ii) Removing studies that had not achieved log normality; (iii) Transformation of log normal to the normal distribution; (iv) Meta-analysis; and (v) Converting the estimates to their original scale.

Prior to conducting the Bayesian hierarchical meta-analysis, we adjusted for differences among the baseline population characteristics of the studies included for analysis. In the original studies, the numbers exposed to PM in each country varied. To avoid considerable disparity across studies, we calculated the premature mortality rate for each respective study year by dividing the country specific number of premature deaths by the population for the same year. Log normality of the mortality rates was assumed and checked using properties of the log normal distribution (Appendix A). Two studies did not satisfy the properties of the log-normal distribution and were excluded from the meta-analysis [18,19]. The transformations between the corresponding log normal and normal distributions were undertaken with the usual conversion equations in conjunction with exploiting properties of the log normal distribution in order to calculate the variances of the log normally distributed mortality rates [20,21], as detailed in Appendix A. The meta-analysis estimates were then transformed back to their original scale.

The analysis was carried out in freeware R, version 2019 [22] using the bayesmeta package version 2019 [17] [https://cran.rproject.org/web/packages/bayesmeta/bayesmeta.pdf] (accessed on 30 January, 2020). The bayesmeta package derives the posterior distributions of the synthesized mean and heterogeneity parameter and their posterior joint distribution. The code used to carry out this analysis is available in Appendix A. The forest plots that were generated from the Bayesian analysis demonstrate the log normal mortality rates with 95% credible and prediction intervals mapped to the normal distribution. Heterogeneity plots were generated to display the posterior joint density of the log normal mortality rate and heterogeneity (τ) parameters, with a darker shading area corresponding to a higher probability density.

We used a non-informative prior as opposed to an informative prior. In the absence of clear prior evidence for mortality rates, we chose this conservative option because non-informative priors have a minimal effect on the analysis [23]. Furthermore, we chose a random effect meta-analysis instead of fixed effect meta-analysis (a more conservative approach) which assumes the potential for the original study samples to arise from different populations. In Bayesian hierarchical meta-analysis, if the number of studies is less than 20, the random effect model is the analysis of choice [23].

## 3. Results

The systematic search revealed 3248 papers. Following the removal of duplicates, 2849 remained for title and abstract screening. Once the title and the abstracts were screened, 309 papers were available for full text review. Of those, 36 published papers met the inclusion criteria and reported estimates on premature mortality (Figure 1). Sixteen of these papers were included in the Bayesian hierarchical meta-analysis.

The 36 published papers came from various countries; six were from India [4,24,25,26,27,28], three from the USA [29,30,31], 14 from China [19,32,33,34,35,36,37,38,39,40,41,42,43,44] and one each from Korea [45], Czech Republic [46], Canada [47], France [48], Yugoslavia [49], Japan [18] and Sweden [50]. There were four global studies [1,51,52,53] and two studies from the Asian region [54,55] (Table 1 and Table 2, Figure 2). Figure 2 shows the geographic distribution of the included studies, with the exception of the global and the Asian region studies.

Although the exposure assessed was PM, the size of the particles differed among the studies. Most studies assessed PM_2.5_ [1,4,24,25,29,30,31,32,33,34,35,37,39,41,42,43,44,45,51,52,53,54,55,56], some measured PM_10_ [19,36,38,40,46], while a number of studies included both [26,27,28,48,49,50]. Only one study specifically mentioned particles between PM_2.5_ and PM_10_ [18] (Table 1) and this paper was not included in any of the meta-analyses.


The techniques used to assess exposures varied among studies. Most studies used spatial modelling, using different modelling techniques, and four studies used satellite-based measures [4,35,43,44] (Table 1 and Table 2). Ten studies directly measured PM levels [18,19,26,28,33,36,39,40,45,49] (Table 1 and Table 2).

The outcome, premature mortality, was calculated based on some existing measurement of the country specific life expectancy. For this outcome, sixteen studies used the Global Burden of Disease data [1,4,24,25,28,33,35,37,38,41,42,43,44,52,53,58] while others used life tables [26], WHO data [51,54] and country statistics [18,19,27,29,30,31,32,34,36,39,40,44,45,46,47,48,49,50] (Table 1 and Table 2). No studies were determined to be of poor quality based on the Newcastle–Ottawa scale.

### 3.1. Studies Not Included in the Bayesian Meta-Analyses

Twenty studies were not included in the meta-analyses. Figure 3 shows the number of publications and the area/country of origin of the studies that were not included in the Bayesian meta-analysis.

Among the studies that were not included in the Bayesian meta-analysis, two publications [27,50] reported results based on both PM_10_ and PM_2.5_, while 13 studies reported [1,4,25,34,35,39,43,44,51,52,53,54,57] only on PM_2.5_ and five [18,19,36,40,46] reported only on PM_10_. (Table 1). The presentation of results was different across the studies; however, the direction of the associations was similar, showing an increase of premature mortality.

### 3.2. Results of the Bayesian Meta-Analysis

The extracted premature mortality rate of the eligible studies was utilized for the meta-analysis (Table 2). Fifteen studies were included in the meta-analysis of PM_2.5_ and three in the meta-analysis of PM_10_, while two studies that investigated the outcome based on both exposures were included in the respective analyses.

The Bayesian hierarchical meta-analysis forest plots report on the stepwise analysis. This approach is hierarchical, which differs to conventional meta-analysis. The first level of the forest plot corresponds to the relevant results of the participants in the study and the second level is generated as the study participants are nested within a study and, here, we assume the sample derived is a randomly selected sample from the exposed population.

Studies included in the PM_2.5_ analysis were published after 2013 and represented a limited number of countries. Six studies were from China, three from USA, three from India with one study each from Canada, Korea and Yugoslavia. City specific information was available only in one study [28].

The analysis based on PM_10_ represented studies from France, China and two cities from India. Therefore, most evidence here came from the Asian continent.

Values in the PM_2.5_ forest plot indicate the log mortality rate mapped to their corresponding normal distribution values. After conversion back to the original scale, the annual estimate of premature mortality due to PM_2.5_ was 253 (95%CI: 90, 643) deaths per 1,000,000 population globally (Figure 4). The predicted value, overarching the sampling error of individual studies, is the expected mean value of a future study which is 269 (95%CI: 15, 3083) per 1,000,000 population.

Similarly, the transformed results of the PM_10_ forest plot (Figure 5) indicate that the annual estimate of premature mortality due to exposure to PM_10_ was 587 (95%CI: 1, 39,746) deaths per 1,000,000 population. However, when the sampling errors of individual studies were removed, the predicted mean result for a future study was 645 (95%CI: 0, 16,106) per 1,000,000 population.

### 3.3. Heterogeneity of the Studies

Figure 6 and Figure 7 illustrate the joint posterior density of heterogeneity τ and the effect µ (log mortality rate), for PM_2.5_ and PM_10_, respectively. The darker area on the plots indicates the area of higher probability density. Red lines represent the 50%, 90%, 95% and 99% credible intervals of the joint distribution. The blue solid line is the conditional posterior mean log mortality rate as a function of heterogeneity, with the blue dashed lines corresponding to the 95% credible interval. The green lines indicate the marginal posterior median and 95% credible intervals for both parameters.

The observed heterogeneity for the pooled studies for PM_2.5_ was 1.06 (95%CI: 0.23, 2.06) and for PM10 it is 1.9 (95%CI: 0.00, 10.50).

When the true heterogeneity is compared between the PM_2.5_ and PM_10_ meta-analyses, the between-study variance (true heterogeneity) was high among the studies that have assessed the outcome based on PM_10_.

## 4. Discussion

In this systematic review, we identified thirty-six studies of either good or fair quality assessing the association between ambient PM_2.5_ and/or PM_10_ and premature mortality. All studies reported a positive association. In the meta-analysis, in which we included sixteen studies, we observed that 253 premature deaths per million population are associated with exposure to ambient PM_2.5_. Prediction estimates indicated that the magnitude of the PM_2.5_—premature mortality relationship will increase in future studies. We obtained unstable estimates for PM_10_, most likely due to the high level of heterogeneity among studies included.

This is the first systematic review and meta-analysis conducted to assess the association between ambient PM (both PM_2.5_ and PM_10_) and premature mortality. Our findings reflect a previous meta-analysis based on 53 studies that explored the association between ambient PM_2.5_ and all-cause mortality, which found that a 1 μg/m^3^ increase in PM_2.5_ was associated with a significant 1.29% increase in all-age all-cause mortality [11]. Similarly, Hanigan and colleagues reported a positive association between anthropogenic PM_2.5_ and premature mortality in Australia [59], albeit not a meta-analysis. A recent systematic review and a meta-analysis conducted by Jie et al. [60] also found that exposure to PM_2.5_ and PM_10_ increases mortality. In addition to mortality studies, studies which assessed Disability Adjusted Life Years (DALYs) and Health Adjusted Life Years (HALYs) as outcomes have also found that PM exposure increases health burden [61]. However, the DALYs does not include years of life lost and the HALYs calculation includes both morbidity and mortality data. In this study we did not include either of these measurements as outcomes thereby enabling us to understand the impact on years of life lost due to premature mortality, which is a long-term exposure to particulate matter pollution. Not including papers with DALYs and HALYs estimates did not bias our findings given only a limited number of papers were excluded.

It is important to highlight that our assessment has only focused on ambient PM. We considered household air pollution as a separate exposure, as in the Global Burden of Disease study. However, household air pollution and ambient air pollution are interlinked exposures as each one contributes to the other. Indeed, it has been found that emissions from the use of unclean fuels for domestic energy, when compared to other emissions such as industry and road traffic, have the largest impact on premature mortality globally [62].

The biological plausibility of the association observed cannot be underestimated. With increasing industrialization and urbanization in most regions, more PM is released into the environment, which has a negative impact on the cardiovascular, cerebrovascular and respiratory systems. This, in turn, increases the risk of mortality before the expected life expectancy. Moreover, the causal relationship we found is supported by many studies. Brook et al. [63] and Pope et al. [64] reported short term changes in PM_2.5_ levels which lead to changes in daily mortality rates. Thurston et al. [65] also highlighted that PM_2.5_ increases IHD (Ischemic Heart Diseases) and mortality and reported a dose response association. Many studies, including the Harvard Six Cities study, have also found that long term exposures to PM_2.5_ (Dockery et al. and Pope et al.) increase mortality and that the overall reduction of PM_2.5_ can reduce the mortality rates, confirming its causal association.

The advantage of our study is the statistical approach used. The Bayesian hierarchical meta-analysis, compared to the conventional meta-analysis, assesses the predicted credible intervals taking the weights of the reference population rather than the individual study results. When compared to a conventional meta-analysis, Bayesian hierarchical methods utilize a prior probability distribution in assessing this. Therefore, Bayesian hierarchical random-effect models can obtain accurate pooling effects, even with a limited number of studies in the meta-analysis. Furthermore, conventional meta-analysis cannot incorporate extreme values and small studies due to the systematic difference, limiting its application to our research question [66]. In contrast to the conventional meta-analysis, Bayesian hierarchical meta-analysis can address these issues [67].

While reading this review, an important point to note is that the strategies undertaken by individual countries to reduce the emission of PM are not uniform across the globe. Therefore, our pooled estimate of premature mortality may vary according to these varying mitigation strategies. The finding of our study is a concern pointing to the urgency of implementing strategies to mitigate this growing environmental risk factor for premature mortality; the impact of ambient PM_2.5_ on premature mortality is remarkably high when considering the current global population and the predicted population growth in the coming decades. Although we observed an increased premature mortality for PM_10_, the confidence interval was extremely wide, indicating an unstable estimate. Results therefore should be interpreted cautiously. The considerable variation observed was most likely due to the heterogeneity among the studies, and future research efforts need to focus on the effects of PM_10_ and premature mortality.

As with all studies, there are a number of limitations. First, heterogeneity among studies may have hidden the real burden of premature mortality due to PM exposure. The studies we analyzed do not represent the global burden of premature mortality due to PM, or the urban rural disparity, as we did not have data representing all countries of the world. Indeed, most of the studies included were conducted in China and India. Although these countries account for 36% of the world’s population, they are also among the most polluted countries, so less polluted countries may be underrepresented in our study. Further, within an individual country, the available data only represents a sample of the population, which may not reflect the true impact. The majority of studies included in our study did not adjust for weather conditions or other associated conditions. The epidemiology-based exposure dose-response functions that were applied, how premature mortality was calculated, and other factors associated with life expectancy may also hide the true association. Second, we were unable to conduct a subgroup analysis, for example by region, due to the limited number of studies and lack of variation in the countries where research was conducted. None of the studies commented on causality rather than association. Third, meteorological effects on particulate matter pollution were not quantified in our analysis. Furthermore, we have excluded the satellite-based studies from our analysis.

Notwithstanding these limitations our approach, namely pooling of the available study results to obtain a summary measure and then to statistically model the reference populations of the included studies using Bayesian hierarchical meta-analysis, is meaningful for analyzing the impact of an environmental exposure(s) in contrast to analyzing a selected sample. This has enabled findings related to environmental exposures, such as PM, where the exposure cannot be confined to a sample population, and a key outcome such as premature mortality. We recommend that future systematic reviews consider this approach when collating evidence on environmental exposures and outcomes.

## 5. Conclusions

Existing evidence indicates a positive association between ambient PM_2.5_ and premature mortality, even while accounting for heterogeneity between studies. Evidence for PM_10_ remains inconsistent. This is one of few meta-analyses that has explored the causal association between PM and premature mortality, taking into account the heterogeneity found in the various reported studies. This study, therefore, strengthens our current knowledge of the important relationship between exposure to PM and health outcomes, highlighting the urgency to mitigate the growing exposure to air pollutants. 

## Figures and Tables

**Figure 1 ijerph-18-07655-f001:**
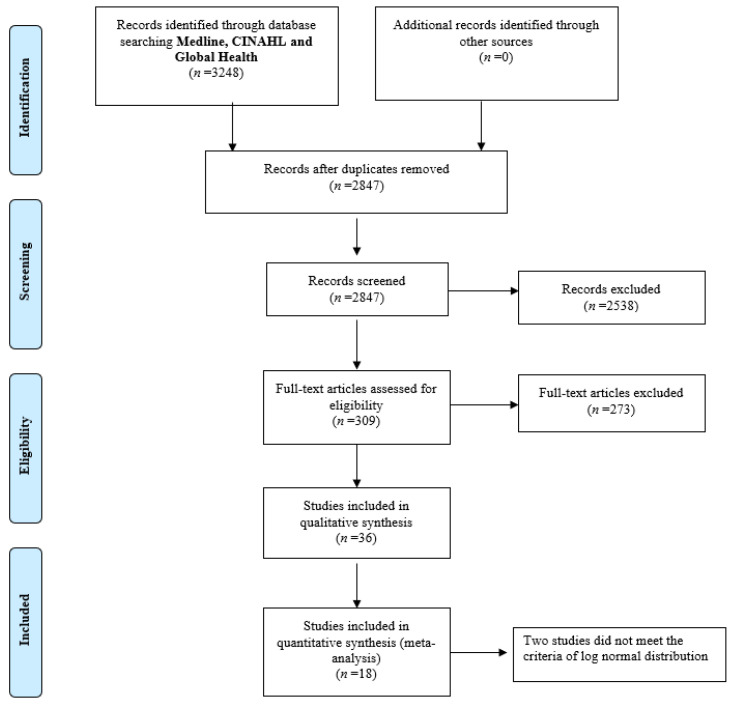
PRISMA flow chart.

**Figure 2 ijerph-18-07655-f002:**
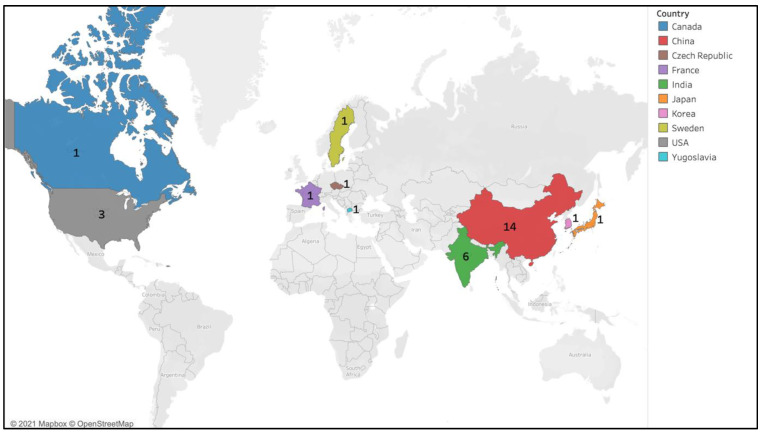
Geographical distribution of the selected studies, indicating the number of publications.

**Figure 3 ijerph-18-07655-f003:**
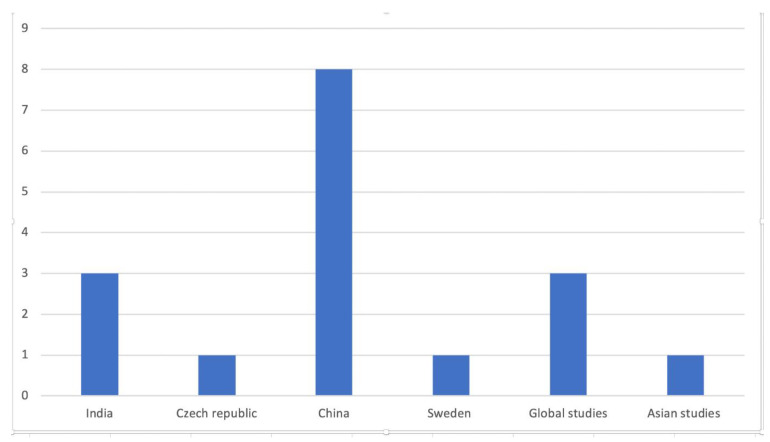
Number of publications that were not included in the meta-analysis based on the area/country.

**Figure 4 ijerph-18-07655-f004:**
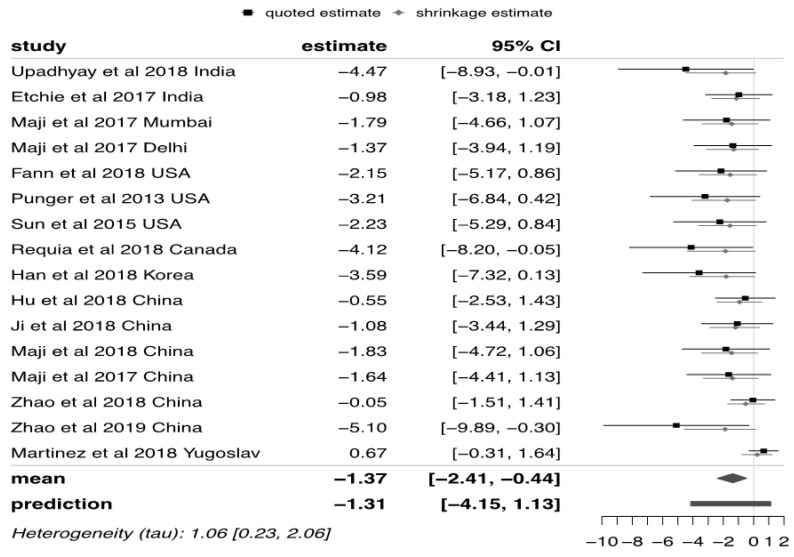
Forest plot PM2.5. The values of the forest plots indicate the log mortality rate mapped to their corresponding normal distribution values.

**Figure 5 ijerph-18-07655-f005:**
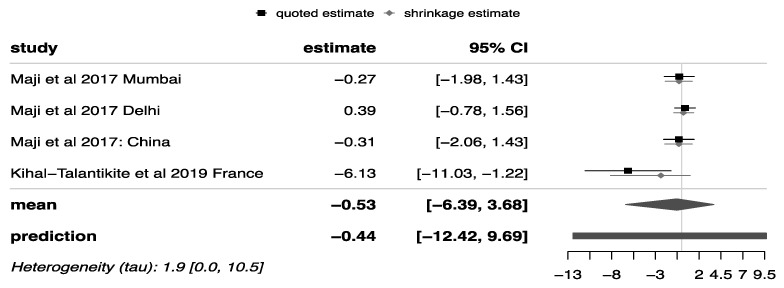
Forest plot PM10. The values of the forest plots indicate the log mortality rate mapped to their corresponding normal distribution values.

**Figure 6 ijerph-18-07655-f006:**
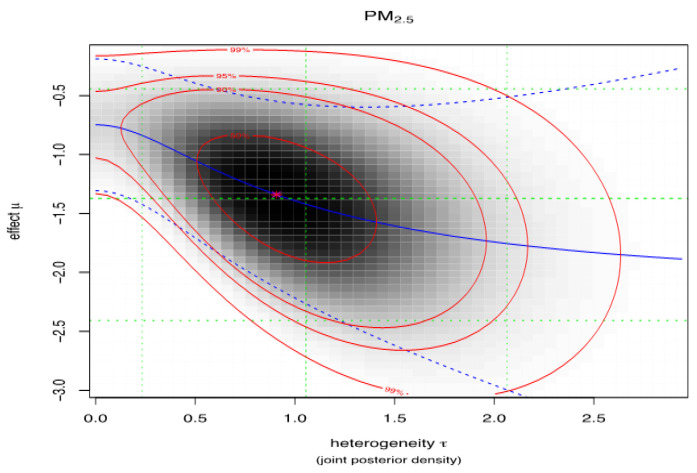
Heterogeneity plot PM_2.5_.

**Figure 7 ijerph-18-07655-f007:**
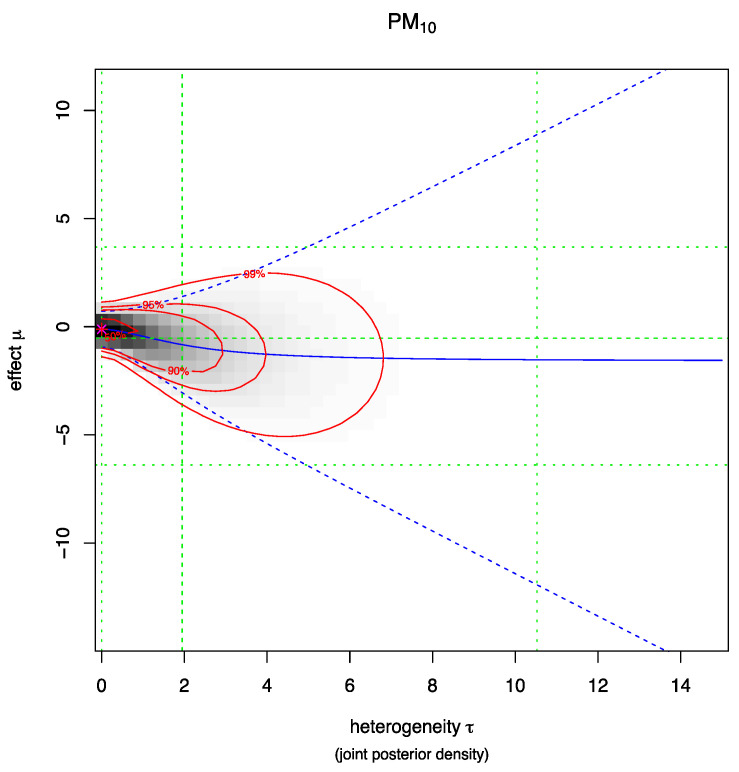
Heterogeneity plot PM_10_.

**Table 1 ijerph-18-07655-t001:** Studies that were not eligible for meta-analysis.

Researcher, Year of the Publication Country	Size of the PM Exposure Ascertained by:	Referred Data to Calculate Premature Mortality:	Results:					Study Quality
Chowdhury 2018 India [25]	PM_2.5_ annual average Estimate up to 2100 by applying changes in PM_2.5_ from baseline period (2001–2005) derived from Coupled Model Inter-comparison Project 5 (CMIP5) models to the satellite-derived baseline PM_2.5_	Global Burden of Disease data	Time		Estimated premature deaths Annual mean for 1,000,000 population			Good
			2031–2040		18.1 ± 4.6			
			2061–2070		10.5 ± 3.5			
			2091–2100		6.5 ± 2.6			
Guttikunda et al., 2012 [27] India Delhi and its satellite cities—Gurgaon, Noida, Greater Noida, Faridabad, and Ghaziabad	PM_2.5_ and PM_10_ Annual average Calculated using Atmospheric Transport Modelling System (ATMoS)	2010 mortality data India	Estimated premature deaths for the year 2010 is between 7350–16,200					Good
Jain et al. 2017 India [4] Holy city Varanasi	PM_2.5_ Annual average Measured using Satellite-retrieved AOD	Global Burden of Disease data	5700 (2800; 7500) annual premature deaths were estimated due to PM_2.5_ (0.16% of the population)					Fair
Buleiko et al. 2017 Czech Republic [46]	PM_10_ annual average Automatic and gravimetric sampling methods	Health Statistic Yearbook data for the country	Year	PM_10_ annual average (SD) Premature deaths: annual (SD)				Good
				T1 (Traffic, Urban, Residential)	T2 (Traffic, Urban, Trade)	B1 (Background, Urban, Residential)	B2 (Background, Urban, Residential, Trade)	
			2009	30.13 ± 8.66 22 ± 16	33.19 ± 15.35 32 ± 21	24.43 ± 5.71 15 ± 12	34.52 ± 8.81 31 ± 14	
			2010	34.33 ± 11.52 29 ± 19	33.84 ± 17.26 48 ± 14	27.00 ± 7.57 22 ± 14	31.43 ± 9.21 24 ± 17	
			2011	30.90 ± 12.28 28 ± 19	30.33 ± 15.92 35 ± 22	26.97 ± 9.70 21 ± 17	29.58 ± 12.74 26 ± 20	
			2012	30.32 ± 8.33 27 ± 14	27.98 ± 13.03 31 ± 17	24.15 ± 4.27 13 ± 9	33.30 ± 9.04 28 ± 16	
			2013	27.29 ± 8.26 27 ± 11	34.87 ± 12.03 35 ± 18	22.48 ± 6.76 19 ± 7	27.13 ± 7.20 22 ± 12	
Li et al. 2018 China [34]	PM_2.5_ annual mean GEOS-Chem chemical transport model by Satellite data	Direct follow-up data	1,765,820 people aged 65 years and older in China in 2010 had premature deaths related to PM_2.5_ exposure					Fair
Lu et al. 2019 China [35]	PM_2.5_ annual satellite-retrieved	Global health data exchange	For the year 2017: 962,900					Fair
Ma et al. 2016 China [36]	PM_10_ annual average Directly measured	China statistical yearbook	2004 to 2013, annual premature deaths attributable to China’s outdoor air pollution ranged from 350,000 to 520,000					Good
Nie et al. 2018 China [39]	PM_2.5_ hourly and daily and annually Directly measured	China Public Health and Family Planning Statistical Yearbook	In 2014, the AFs (%) for COPD, LC, IHD, and stroke were 23% (95% CI: 12, 32%), 29% (95% CI: 11, 40%), 30% (95% CI: 21, 48%), and 46% (95% CI: 17, 57%), respectively. In 2015, with the decrease of PM_2.5_, the AFs had fallen to 20% (95% CI: 10, 29%), 25% (95% CI: 8, 35%), 28% (95% CI: 19, 44%), and 44% (95% CI: 15, 55%).					Good
Zhao et al. 2016 China [40]	PM_10_ Directly measured daily calculated for the year	Health statistic yearbook	Air pollutant	Disease causing premature deaths		Dose response coefficient		Fair
			PM_10_	Respiratory disease		0.0048		
				Cardiovascular diseases		0.0019		
Xie et al. 2016 China [43]	PM_2.5_ Satellite derived analysis	Global Burden of Disease data 2000–2010	In total 1.25 million premature deaths due to anthropogenic PM_2.5_ in 2010					Fair
Wang et al. 2018 China [44]	PM_2.5_ annual average Satellite derived analysis	Provincial level data and global burden of disease data	Premature deaths attributed to PM_2.5_ nationwide amounted to 1.27 million in total					Fair
Nawahda et al. 2013 Japan [18]	PM_7.5–10_ Directly monitored by the National Institute of Environmental studies	Japan Statistics Bureau	2006–2009 total of 40,000 premature deaths attributed In 2009: 8347 (95%CI: 2087, 16,695)					Good
Huang et al. 2011 China [19] Pearl River	PM_10_ annual average Directly measured by Environmental monitoring center	Health Statistic Yearbook data 5.71 × 10^7^			Mean (95%CI)			Good
			Acute PM_10_ effect		12,786 (3449, 20,837)			
			Chronic PM_10_ effect		15 (4, 26)			
Segersson et al. 2017 [50] Sweden	PM_2.5_ and PM_10_ annual mean dispersion modelling to assess annual mean exposure	Swedish Cause of Death Register	Number of premature deaths: PM_2.5_: 256 PM_2.5–10_: 54					Good
Fang et al. 2013 Global [51]	PM_2.5_ modelled annually Using AM3 design	WHO data	Global estimate over 21st century annually (accounts for climate change): 100,000 95%CI: (95% CI: 66,000, 130,000)					Good
Wang et al. 2017 Global [1]	PM_2.5_ annually CMAQ modelling	Global Burden of Disease data	PM_2.5_-mortalities in East Asia and South Asia increased by 21% and 85% respectively, from 866,000 and 578,000 in 1990, to 1,048,000 and 1,068,000 in 2010. PM_2.5_-mortalities in developed regions (i.e., Europe and high-income North America) decreased substantially by 67% and 58% respectively					Good
Silva et al. 2016 Global [52]	PM_2.5_ Annually Integrated exposure–response model	Global Burden of Disease data	2.23 (95% CI: 1.04; 3.33) million premature mortalities/year in 2005					Good
Silva et al. 2016 Global [53]	PM_2.5_ Annually to forecast ACCMIP models	Global Burden of Disease data	2030: 17,200 (95%CI: −386,000, 661,000) 2050: −1,210,000 (95%CI: −1,730,000, −835,000) 2100: −1,310,000 (95%CI: −2,040,000, −174,000)					Good
Nawahda et al. 2012 [54] South East Asia	PM_2.5_ annually CMAQ modelling	WHO data	2000: 237,665 (95%CI: 59, 416,475) 2005: 405,035 (95%CI: 101,259, 810,070) 2020: 313,438 (95%CI: 78,360, 626,876)					Good
Shi et al. 2018 [57] South and South East Asia	PM_2.5_ Annual GEOS-Chem chemical transport model	Global Burden of Disease data	During 1999–2014, the estimated total average annual premature deaths mortality due to PM_2.5_ exposure in SSEA reached 1,447,000 (95% CI: 9,353,00l, 2,541,100)					Good

**Table 2 ijerph-18-07655-t002:** Studies Included in the Bayesian Hierarchal meta-analysis.

Researcher, Year of the Publication Country	Size of the PM Exposure Ascertained by:	Referred Data to Calculate Premature Mortality and the Baseline Population:	Results:		Quality of the Study:
Upadhyay et al., 2018 [24] India	PM_2.5_ annual average Calculated using WRF-Chem simulation	Global Burden of Disease data and Indian census data 1.23 × 10^9^	PM_2.5_ level µg m^−3^	Number of premature deaths avoided annually if completely mitigated	Good
			Transport: 3.8 ± 4.3 Industrial: 5.5 ± 2.7 Energy: 2.2 ± 2.3	92,380 (95%CI: 40,978, 140,741)	
			Residential: 26.2 ± 12.5	378,295 (95%CI: 175,002, 575,293)	
			Pooled estimate: 187,400 (95%CI: 47,073;746,038) premature deaths annually if completely mitigated the effect of PM_2.5_ annually		
Etchie et al. 2017 India [26] Nagpur city	PM_2.5_ & PM_10_ Annual average Directly measured	Life tables 4.65 × 10^6^	Premature deaths in 2013 (95%CI) due to PM_2.5_ was 3300 (2600, 4200) Population in Nagpur is 4,653,570		Good
Maji et al. 2017 India [28] Mumbai and Delhi	PM_2.5_ and PM_10_ annualDirectly measured if unavailable in some stations a conversion factor was used	Global Burden of Disease data Mumbai: 2.25 × 10^7^ Delhi: 1.82 × 10^7^	The annual average deaths attribute to PM_2.5_ in Mumbai and Delhi was 10,880 (95%CI: 5520, 16,387) and 10,900 (95%CI: 6118, 15,879). Annual average premature deaths attributable to PM_10_ was around 25,006 (95%CI: 16,550; 32,346) and 32,115 (95%CI: 22,619; 39,192) for year 1991–2015 in the urban area of Mumbai and Delhi.		Good
Fann et al. 2018 USA [29]	PM_2.5_ annual average CMAQ modelling	BenMAP-CE software (USA Environmental protection agency. Washington, DC, USA) Using country level data 3.18 × 10^8^	Year	Number of premature deaths and 95%CI	Good
			2005	150,000 (100,000, 200,000)	
			2011	124,000 (84,000, 160,000)	
			2014	121,000 (83,000, 160,000)	
Punger et al. 2013 USA [30]	PM_2.5_ annual average CMAQ modelling	BenMAP Based on centre for Disease Control Data 2.95 × 10^8^	66,000 (95%CI: 39,300; 84,500) premature deaths in 2005		Good
Sun et al. 2015 USA [31]	PM_2.5_ annual WRF/CMAQ modelling	BenMAP-CE software Using country level data 2.82 × 10^8^	103,300 (70,400; 135,700) for the year 2000 60,700 (35,000; 86,000) for the year 2050		Good
Requia et al. 2018 Canada [47] Hamilton	PM_2.5_ annual estimates EPA’s MOVES model	Statistics Canada 5.19 × 10^5^	Total premature deaths over Hamilton to be 73.10 (95%CI: 39.05; 82.11) deaths per year.		Good
Kihal-Talantikite et al., 2018 [48] France	PM_2.5_ and PM_10_ The ESMERALDA Atmospheric Modelling system	Paris Death Registry	2007–2009, the number of attributable deaths was equal 3209 (95%CI: 1938, 3355) and 2662 (95% CI: 2859, 3553)		Good
Han et al. 2018 Korea [45]	PM_2.5_ annual average Directly measured CMAQ method	Using population census data 5.10 × 10^7^	In 2015 the number of premature deaths due to PM_2.5_: 8539 (8428; 8649)		Good
Hu et al. 2018 China [32]	PM_2.5_ annual average Mean exposure taken from average from 60 cities CMAQ model	China Public Health and Family Planning Statistical Yearbook 2014 1.35 × 10^9^	In 2013 PM_2.5_ related premature deaths for adults ≥30 years old is approximately 1.30 million, 95%CI: 0.69l, 1.78 million		Good
Ji et al. 2019 China [33] Beijing-Tianjin-Heibei	PM_2.5_ Directly measuredModelled with previous data	Global Burden of Disease data 1.05 × 10^8^	74,000 (95% confidence interval CI: 43,000, 111,000) premature deaths were attributable to PM_2.5_ exposure in 2013.		Good
Maji et al. 2018 China [37]	PM_2.5_ Air quality monitoring network measurements	Global burden of disease data 1.37 × 10^9^	PM_2.5_ in 161 cities was 652 thousand (95%CI:298, 902) thousand premature deaths in 2015		Good
Maji et al. 2017 China [38]	PM_2.5_ and _10_ Air quality monitoring network	Global Burden of disease data 1.37 × 10^9^	Total premature deaths in China from 2014–2015 PM_2.5_ 722,370 (95%CI: 322,716, 987,519 PM_10_ pollution has caused 1,491,774 (95%CI: 972,770, 1,960,303) premature deaths (age > 30) in China		Good
Zhao et al. 2018 China [41]	PM_2.5_ annual average CMAQ modelling	Global Burden of Disease Data 1.37 × 10^9^	PM_2.5_ related premature deaths in 2005 amounted to 1.72 (95%CI: 1.47, 1.99) million. The marginal contribution of household fuels was estimated at 0.91 (0.72, 1.13) million, 53% (46, 60%) of the total		Good
Zhao et al. 2019 China [42] Beijing, Tianjin, Hebei	PM_2.5_ meteorologically assessed CMAQ modelling	Global Burden of Disease data 1.12 × 10^8^	Exposure:long term PM_2.5_		Good
			COPD	17.42(95%CI: 9.45, 24.40) thousand	
			IHD	36.29(95%CI: 27.24, 48.48) thousand	
			Lung cancer	13.53(95%CI: 5.19, 18.19) thousand	
			Stroke	61.91(95%CI: 27.71, 79.93) thousand	
			Acute lower respiratory infection	0.91(95%CI: 0.62, 1.14) thousand	
			Annual premature deaths: Short term PM2.5 18.7 thousand Long term PM2.5 130.1 thousand		
Martinez et al. 2018 Yugoslav Republic of Macedonia [49]	PM_2.5_ and PM_10_ annual average Directly measured	State statistical office 5.44 × 10^5^	PM_2.5_: 1199 premature deaths (95%CI: 821, 1519) in the year 2012		Good

## Data Availability

Not applicable.

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
