# Peer review of "Particulate Matter and Premature Mortality: A Bayesian Meta-Analysis"

_ijerph, 2021, doi:10.3390/ijerph18147655_

Round 1

Reviewer 1 Report

This manuscript covers an interesting topic which is suitable for IJERPH, regarding its aims and scopes. The authors present a paper which has mixed features, so that it might be consider partly a review paper and partly a research paper. Nonetheless, in the current form, I do not think it fits the standards of quality required for publication in IJERPH. There are several critical issues which should be fixed prior to publication:

1. Methods.
- At some point, the methods and criteria to include or not a specific study in the analysis performed in this research are somehow confusing. The authors are encouraged to clarify the criteria from the very beginning. Otherwise, it may seem to be an ad-hoc decision.
- There is a lack of detailed description of the bayesian analysis specifically performed. Rather than the link to the software used, I consider an average reader of IJERPH would appreciate a description of the steps followed, and a clear explanation of what you want to test. Furthermore, including significant references of studies with comparable methodology might be useful.

2. Results, Discussion and Conclusions.
- A limited interpretation of the results is presented. My feeling is that there is a failure to exploit all the data collected. Still much work of analysis shall be performed in order not to dilute the good job carried out in the first part of the manuscript, namely finding and classifying the papers.
- The plots information is typically not totally extracted, and a deeper analysis on them is required.
- A main conclusion is presented in lines 252-257. However it is something previously pointed out by many researchers. The information in lines 265-283 seems to be much more methodology than actual discussion. Anyway I would expect these last two sections to be a bit sharper. Particularly the section of Conclusions is extremely short.

3. Concision, style and formal presentation.
- The abstract is way too long, and not concise at all. It is hard to understand the goal of the paper and the results obtained.
- Occasionally, many ideas are mixed together in the same paragraph, but not really connected to each other. There are too many words with extra unnecessary hyphens. Besides, the use of commas is sometimes not appropriate and therefore it can hamper reading the manuscript.
- There are some paragraphs whose interest is questionable. Is it really relevant to include, for instance, the content of lines 183-188? In addition to this, I find superfluous the lines 258-264.
- Some sections appear to be incomplete. An example in case is the Authors Contributions.

As a whole, I recommend to reorganize the manuscript taking into account the previous points, and maybe to reformulate the way the authors present the paper, considering either one of these two options:
- a deeper statistical analysis
- a pure review paper, extracting the main findings of all (or most of) the papers selected and their techniques

As the results in this version presented are promising, but still lack of a stronger conclusion, I find the first one the most interesting option, that is, to deep into the statistical analysis. I suggest to test the values of the expected mean for a future study (i.e., for PM2.5, 269/1000000 population). This can be done with one or several external studies, which may have been performed after all the works here considered.

Author Response

Attached is the reply to reviewer one.

Reviewer 2 Report

The authors have discussed A very hot topic “particulate matter and mortality”. The design and methodology seems unique in this context. Hence, I
would like to highly recommend submitted research manuscript for possible publication. While
the paper is nicely written, some minor areas of improvements are as follows:
• Abbreviations should be carefully checked and given at the start.
• There are spelling errors and language issues detected.
• Kindly prepare some good figures to represent the research.
• Few latest references from the journal on particulate matter and mortality may be included.
• English editing is needed in some parts of the manuscript.
• Several sentences are presented without references try to use good published paper
to improve the novelty of your work.
• Use one citation style instead of variation in it.

Author Response

Please see the attachment for the reply of reviewer's comments.

Reviewer 3 Report

There are few reviews to assess the potential association between particulate matter in the environment and premature mortality adjusted for population differences. That adds a pinch of originality to it. However, more clarity should be provided as to how heterogeneity among studies have hidden the burden of premature mortality due to the exposure of PM.

The present review uses a Bayesian Hierarchical meta-analysis. And a prerequisite for it is the normality of the data. In order to achieve normality, the log normal rates have been transformed to their values in the corresponding normal distribution with the conversion equations. Further information on the studies that did not comply with this criterion should be provided. 

The conclusions assert that PM cause premature mortality. But more evidence is needed. Further discussion should be provided on the relation of causality or association.

Author Response

(The authors gave the same response as above.)
